# Lessons from Using Genetically Engineered Mouse Models of MYC-Induced Lymphoma

**DOI:** 10.3390/cells12010037

**Published:** 2022-12-22

**Authors:** René Winkler, Eva-Maria Piskor, Christian Kosan

**Affiliations:** Department of Biochemistry, Center for Molecular Biomedicine (CMB), Friedrich Schiller University Jena, 07745 Jena, Germany

**Keywords:** MYC, B-cells, lymphoma, Eµ-Myc transgene, genetically engineered mouse models

## Abstract

Oncogenic overexpression of MYC leads to the fatal deregulation of signaling pathways, cellular metabolism, and cell growth. *MYC* rearrangements are found frequently among non-Hodgkin B-cell lymphomas enforcing MYC overexpression. Genetically engineered mouse models (GEMMs) were developed to understand MYC-induced B-cell lymphomagenesis. Here, we highlight the advantages of using Eµ-Myc transgenic mice. We thoroughly compiled the available literature to discuss common challenges when using such mouse models. Furthermore, we give an overview of pathways affected by MYC based on knowledge gained from the use of GEMMs. We identified top regulators of MYC-induced lymphomagenesis, including some candidates that are not pharmacologically targeted yet.

## 1. Introduction

As observed in most human malignancies, overexpression of MYC leads to fatal cellular metabolism, growth, and signaling deregulation, which defines MYC as a classical oncogene [1,2]. *MYC* rearrangements, copy number amplifications, or mutations are frequently found among non-Hodgkin B-cell lymphomas (B-NHL) and enforce MYC overexpression [3]. More specifically, roughly 80% of Burkitt’s lymphoma (BL), 15% of diffuse large B-cell lymphoma (DLBCL), and 2% of follicular lymphoma (FL) are characterized by *MYC* translocations [3,4,5].

Despite the progress made in the past decades, cancer heterogeneity, insufficient therapy response, and relapse are still risks for patients. As directly targeting MYC with chemical compounds is challenging [6], exploring genetic vulnerabilities in MYC-induced B-cell lymphoma is expected to uncover new “attack points” for cancer treatment.

This review highlights the advantages of genetically engineered mouse models (GEMMs) to understand MYC-induced lymphomagenesis. We focus on the widely used Eµ-Myc transgenic mouse model and discuss the most common challenges when using this model system. We give here an overview of GEMMs of MYC-induced B-cell lymphoma, revisiting over 170 GEMMs with distinct genetic alterations. By thoroughly compiling these data, we were able to identify top regulators of MYC-induced lymphomagenesis, including some candidates that are not pharmacologically targeted yet. At last, we discuss future directions in using GEMMs in B-cell lymphoma research.

## 2. MYC and the Origin of the Eµ-Myc Mouse Model

The transcription factor (TF) MYC is one of the most prominent proto-oncogenes and is composed of a basic helix–loop–helix (bHLH) motif, followed by a leucine zipper that enables its DNA binding and five MYC homology boxes (0-IV), which facilitate stability, protein–protein interactions, and transcriptional regulation [7,8]. Moreover, MYC harbors a nuclear localization signal (NLS) to ensure nuclear import [9]. Here, MYC binds to defined areas in the chromatin. The consensus sequence of the enhancer (E)-box region bound by MYC is characterized by a CACGTG motif and can be found in one-third of MYC binding loci in human B-cells [10]. However, non-canonical E-box (CACATG) and E-box-independent mechanisms contribute to the binding of MYC to roughly 5000 promoter sites in B-cell lymphoma [10,11].

In murine naïve B-cells, MYC was associated with regulatory elements pre-existing in a poised transcriptional state, which is crucial for fast responses toward immunological stimuli [12]. Here, MYC and its interaction with key transcription factors of B-cell identity are essential for cell cycle entry of germinal center (GC) B-cells and memory B-cell formation [13,14,15]. To achieve elevated mRNA translation, MYC regulates the expression of genes encoding transfer RNAs, ribosomal RNAs, and components of the spliceosome [12,16,17,18]. In line with this, murine MYC-deficient B-cells fail to amplify RNA synthesis after stimulation with lipopolysaccharide (LPS) due to impaired chromatin reorganization, which impedes TF occupancy at key promoters [19]. Taken together, physiological MYC is a crucial regulator of transcription and translation in normal cells and a potent driver of B-cell lymphomagenesis in mice and humans (Figure 1).

In 1985, Adams and colleagues constructed seven different transgenic mouse strains by inserting the gene encoding MYC into distinct regulatory regions [23]. From these, constitutive expression of *Myc* under the immunoglobulin enhancers Eκ or Eµ led to lymphoma development originating from B-cells with dramatically increased incidence [23]. Eµ-Myc transgenic mice possess three *Myc* transgenes on chromosome 19 and two somatic *Myc* copies on chromosome 15 and are usually bred in a heterozygous manner [24,25]. If all copies are expressed, or some are silenced similar to BL cell lines, is not clear [26,27]. However, the onset of lymphoma is dose-dependent, as homozygous crossings of Eµ-Myc transgenic mice show dramatically reduced survival [28].

Deregulation of *Myc* transcription in Eµ-Myc transgenic mice occurs stepwise exceeding physiological levels of MYC in pre-tumor cells [12,29,30]. The high abundance of MYC leads to its accumulation at active promoters and enhancers, causing transcriptional amplification by recruitment of factors that phosphorylate RNA polymerase II at serine 2 to speed up transcription, also known as the “general amplifier hypothesis” [16]. Moreover, oncogenic MYC invades low-affinity E-boxes and target sites without E-box motifs [16,31], which can be explained by the biophysical properties of dimeric bHLH domains to non-specifically bind DNA [32].

The “general amplifier hypothesis” is complemented by selective transcriptional regulation of key target genes, generating fatal feedback loops that shape gene expression [12]. These complex pathophysiological mechanisms explain why the expression of an identified core target gene signature for MYC comprising 51 genes could not sufficiently explain how MYC transforms healthy B-cells [33]. In line with the famous “two-hit” hypothesis for tumor development [34], overexpression of MYC in B-cells would generally induce apoptosis or senescence, making the acquisition of secondary mutations necessary for oncogenic transformation [35,36].

At the cellular level, B-cell-restricted overexpression of *Myc* promotes the accumulation of IgM-negative B-cells in the bone marrow [37]. Moreover, peripheral blood from Eµ-Myc transgenic mice allows tracking of large B220^low^ cells over time, correlating with disease onset and progression [28,38]. Evidence exists that B220^low^ cells are the actual pre-tumorigenic cell type, as they are also found in spleens of Eµ-Myc transgenic mice [29,38]. In wild-type mice, B220^low^ cells are mainly found in the bone marrow but not in the spleen, representing either immature B- or plasma cells [39,40].

The median survival of Eµ-Myc transgenic mice was initially determined to be 11 weeks [23], which enables fast survival analysis of cohorts. Diseased Eµ-Myc transgenic mice develop massively enlarged lymph nodes at inguinal, brachial, and cervical sites, splenomegaly, and sometimes thymoma or bowel obstruction due to tumor masses in the abdomen; this was initially described as “multicentric lymphosarcoma with associated leukemia” [23,41]. Later descriptions of diseased Eµ-Myc transgenic mice included behavioral alterations such as “inactivity, lack of grooming, and/or cachexia” [28], which can now be monitored automatically using body temperature, weight, and food and water intake [42]. Cachexia or “wasting syndrome” can occur in Eµ-Myc transgenic mice without visible lymphoma formation, but with significant weight loss [43]. Taken together, the Eµ-Myc mouse model is a useful system for studying B-cell lymphomagenesis and tumor growth, as molecular changes result in phenotypic alterations.

## 3. B-Cell Lymphomas from Eµ-Myc Transgenic Mice Arise in a Competent Immune System and Are Highly Heterogeneous

Eµ-Myc transgenic mice possess a competent immune system with all innate and adaptive immune cell types [36,38,44]. The singular deletion of α/β T-cells and natural killer (NK) T-cells or γ/δ T-cells, which are known for their anti-tumor actions, did not affect the survival of Eµ-Myc transgenic mice [45]. This appears to be the case for lymphomas from Eµ-Myc transgenic mice with mutations of the tumor suppressor p53 [36]. In contrast, lymphoma cells overexpressing BCL-2 were immunologically visible and removed by CD8^+^ T and NK cells [36]. Therefore, this mouse model might be helpful in studying the role of immune cells in counteracting malignant transformation.

The competent immune system enables applications from the growing field of immuno-oncology, including checkpoint inhibitors, cancer vaccines, CAR-T cells, or oncolytic viruses to encounter cancer. Surface expression of PD-1 receptors was found to be increased on cytotoxic T-cells in lymphoma-bearing Eµ-Myc transgenic mice [46], while B-cells from Eµ-Myc transgenic mice showed an upregulation of the inhibitory receptor PD-L1 to prevent T-cell-mediated elimination [25]. The use of chemotherapy and CAR-T-cells against CD19 increased the survival of Eµ-Myc lymphoma cell xenografts, although normal B-cells were diminished as well [47]. Another approach is the “vaccination” of wild-type mice with α-galactosylceramide-loaded (a synthetic CD1d-dependent NK T-cell ligand) and irradiated Eµ-Myc lymphoma cells to lower tumor burden and increase survival after transplantation of malignant cells from Eµ-Myc transgenic mice [48]. A combination of this “vaccine” and an antibody against the immune checkpoint receptor 4–1BB (CD137) resulted in long-term protective effects [49]. Therefore, Eµ-Myc transgenic mice are valuable tools for evaluating potential therapeutic approaches involving the training of the immune system to detect and erase malignant cells.

Lymphomas arising in Eµ-Myc transgenic mice show differences compared to IgM-positive human BL, as constitutive Myc expression occurs from the earliest B-cell progenitor on. Therefore, disease does not necessarily involve GC formation, and all stages of B-cell development can form the tumor cell of origin [36,50]. Mixed phenotypes might be related to multiple events of malignant transformation or cellular de-differentiation rather than “active” differentiation of malignant cells. This is in line with the idea that MYC promotes proliferation but inhibits differentiation [20]. When a specific gene knock-out (KO) blocks or impairs normal B-lymphopoiesis, combination with the Eµ-Myc transgene results in an expansion of the affected B-cell subpopulation. Examples include KO of Msh2 (encoding DNA damage repair protein MSH2) or Prkaa1 (encoding AMP-activated protein kinase AMPKα1) resulting in merely pro-B or pre-B lymphomas [45,51] or, on the contrary, KO of Ube3a (encoding Ubiquitin-Protein-Ligase E3A), which shifts the phenotype towards mature B-cell lymphomas [52].

There is an ongoing debate about using mouse experiments to model human diseases and therapeutic approaches. It became clear in recent years that the experimental design, the mouse model, and the characterization of the patient cohort are critical to obtain transferable data from mouse studies. Eµ-Myc-derived lymphomas are very heterogenous; for example, a study described that individual B-cell lymphomas shared only a quarter of all differentially expressed genes [12]. One would expect that Eµ-Myc lymphomas are similar to human BL due to the Eµ-Myc transgene. However, gene expression signatures close to human DLBCL, including the germinal center-like and activated B-cell-like subtype, were also identified [53]. As DLBCL can be driven by BCL6 translocations or hyperactive RAS signaling [54], the appearance of frequent mutations in *Kras* or *Bcor*, which encodes a repressor of BCL-6, as tertiary drivers in Eµ-Myc lymphomas might explain DLBCL-like gene expression signatures [24].

Transgenic mouse models have been widely used to study signaling pathways and test therapeutic agents to understand oncogenic mechanisms. For example, the deregulated mTORC1 signaling in Eμ-Myc lymphomagenesis can be used by treatment with the mTORC inhibitor everolimus, even at the pre-malignant stage, to delay lymphoma onset [55]. These findings from mouse experiments have been validated in human B-cell lymphoma later, including BL and DLBCL, and led to promising clinical trials [56,57,58].

A recent study showed that lymphomas from Eµ-Myc transgenic mice exposed to chemotherapy recapitulate the gene expression profiles of patients suffering from DLBCL. The authors identified a senescence-associated gene signature, termed “SUVARness”, and elevated H3K9me3 marks that predict a favorable outcome in DLBCL patients [59].

These examples clearly show that drug screening in mouse models can be successfully translated into human therapies and that the Eµ-Myc mouse model is a valuable tool for studying human B-cell lymphomas due to similarities in genetic alterations.

## 4. Critical Genes for MYC-Induced Lymphomagenesis Share Common Pathways

More than 300 peer-reviewed studies involving Eµ-Myc transgenic mice have been published (Figure 2A). We identified 172 different GEMMs presented with survival curves based on the Eµ-Myc transgene (Table 1). These models included additional genetic perturbations, such as complete KOs, conditional KOs, or overexpression by transgene insertion (Figure 2B). A minority of research studies worked with specific (point) mutations or xenografts created with fetal liver or lymphoma cells (Figure 2B).

Roughly 60% of studies analyzed the survival of Eµ-Myc transgenic mice bred to full KOs, resulting in the deletion of the respective gene throughout the entire body. However, this can disturb immuno-surveillance, angiogenesis, or metabolism by affecting other cells. In contrast, only 15% of the studies were based on cell type-specific KOs using B-cell-specific Cre recombinases, such as CD19-cre (active in pro B-cells) or Mb1-cre (active in pre-pro B-cells) to eliminate the target gene flanked by loxP sites [60,61]. This seems favorable for investigating B-cell lymphomagenesis, but a careful interpretation is needed when normal B-cell development is perturbed by the KO, as a decreased B-cell population stochastically lowers malignant transformation events. Moreover, escaping from genetic inactivation can occur under certain conditions during lymphomagenesis, skewing the obtained survival curves [62]. A more sophisticated approach would be the combination of the Eµ-Myc transgene with Cre recombinases that delete in germinal center B-cells, such as AID-cre, CD21-cre, or Cγ1-cre to study mature B-cell lymphomas with the respective genetic deletion [63,64,65].

To rule out strain- or model-dependent effects, when comparing the survival of all these different studies, a percentage normalized to the used control cohort was calculated. By considering all genes where deletion or overexpression had a significant impact on mouse survival (either <50% reduction or >200% increase in life span), we identified 48 critical genes for MYC-induced lymphomagenesis (Figure 2C,D). This also implies that more than two-thirds of all studies did not observe an effect on survival, as defined by our criteria. Common biological functions among the 48 critical genes were identified, being “regulation of transcription by RNA polymerase II”, “chromatin organization”, “histone modification”, and “regulation of catabolic processes” based on the gene ontology (GO) terms (Figure 3). In addition, “signal transduction by p53”, “apoptotic signaling pathway”, and “DNA damage response” were found as common biological pathways among these critical proteins. The contribution of these pathways in MYC-induced B-cell lymphomagenesis will be discussed in detail in this chapter.

### 4.1. Epigenetic Modifiers Cooperate with MYC to Maintain Uncontrolled Proliferation

Three distinct fail-safe mechanisms prevent cells from becoming cancer: (1) Induction of cell cycle arrest via p21 through activated p53, (2) sequestering of cell cycle promoting factors such as E2F proteins through pRB (retinoblastoma protein), and (3) repression of anti-apoptotic factors or activation of pro-apoptotic factors to induce apoptosis.

GEMMs with B-cell-specific MYC overexpression and deletions or mutations in the p53/ARF/MDM2-axis uniformly showed reduced survival, as expected [69,70,71,72,73,74]. Of note, whole-body KO of Trp53 (encoding p53) can result in T-cell lymphomas, and investigators should be careful to delineate the distinct effects [75]. Interestingly, B-cell-specific KO of Trp53 with Mb1-cre resulted in the development of B-cell lymphomas harboring oncogenic translocations, including Igh/Myc—very similar to Eµ-Myc transgenic mice—and a median survival of 28 days [76]. Lymphomas were also formed when Trp53 was knocked out using CD21-Cre, although these tumors lacked elevated MYC expression [77]. On the contrary, deleting one allele of Rb barely altered the survival properties of Eµ-Myc transgenic mice [71], while a critical role was attributed to levels of E2F in MYC-induced lymphomagenesis [78].

The pRB/E2F-axis is controlled through the interplay of cyclins and cyclin-dependent kinases (CDKs) [79]. MYC directly activates the expression of the gene encoding Cyclin D2, which is essential for driving cell proliferation [80]. RAS, a commonly found tertiary driver in Eµ-Myc lymphomas [24], can activate Cyclin D1 [81]. Cyclin E, for example, is positively regulated by the RNA helicase DDX3, encoded by the sex chromosomes [82]. Tumor formation was heavily impaired in females but not males lacking the X-linked allele Ddx3x using a B-cell-specific KO model crossed to Eµ-Myc transgenic mice [62].

Epigenetic modifiers were shown to be highly involved in impacting the pRB/E2F-axis in Eµ-Myc transgenic mice. This includes the histone acetyltransferase GCN5 (Kat2a), whose homozygous loss resulted in a decreased expression of the genes encoding E2F and Cyclin D and improved survival by almost three-fold [83]. Similarly, heterozygous loss of the gene encoding the histone acetyltransferase MOZ (Kat6a) slowed down the proliferation of malignant B-cells, and pharmacological inhibition of MOZ reduced E2F2 transcription, stopping tumor growth in vivo [84,85].

Cyclin D1 expression was furthermore shown to be regulated by the methyl transferase EZH2 [86]. More specifically, a gain-of-function mutant of EZH2 (Y641F), which is also found among B-cell lymphoma patients [54,87], acted in concert with wild-type EZH2 to elevate activating H3K27 trimethylation marks in Eµ-Myc transgenic mice, resulting in enforced B-cell receptor signaling and lethality [86].

In contrast, the histone methyltransferase SUV39H1 creates repressive histone marks by tri-methylation of H3K9 and directly interacts with pRB, making it crucial for the repression of genes encoding cyclins [88,89]. Therefore, the loss of Suv39h1 alleles reduced the lifespan of Eµ-Myc transgenic mice by 50%, also due to defects in senescence induction in cancer cells [90].

These molecular interactions show how MYC sustains proliferation and overrides cell cycle checkpoints to circumvent cell-intrinsic safeguard programs. Interestingly, the latest subclassification of DLBCL contains a cluster of human B-cell lymphomas, driven by deregulated methyl transferases, such as EZH2 or MLL4 [54,87], that often occur together with MYC and BCL-2 alterations. Many “epi-drugs” exist that target epigenetic regulators in B-NHL [91], which could be further utilized in therapy for MYC-dependent lymphoma or lymphomas with MYC as the secondary driver.

### 4.2. Impairing Direct MYC Interaction Partners Is Most Effective in Prolonging Survival

MYC has many interaction partners (Figure 4), constantly interchanging through competitive binding. MAX (MYC-associated factor X), for example, dimerizes with MYC under physiological conditions at E-boxes through bHLH domains to stimulate the transcription of target genes [7]. The MYC:MAX interaction is crucial for target gene transcription and high MYC levels, as MAX-deficient B-cells had unstable MYC, and MAX-deficient Eµ-Myc transgenic mice significantly extended survival [92]. Therefore, the small molecule inhibitor 10058-F4, which blocks dimerization between MYC and MAX, was expected to be a breakthrough in targeting MYC-dependent cancer [93]. However, 10058-F4 did not achieve adequate results due to high turnover and the demand for high molar concentrations in vivo [6]. New inhibitors that disrupt the MYC:MAX interaction are MYCi975 and MYCMI-7, and they obtained promising results [94,95].

Binding partners of MYC are functional mediators but also might regulate protein stability (Figure 4). The half-life of MYC is the oncogene’s weak spot, as half of all MYC must be synthesized newly every 30 min, and emphasizes its strong regulation [96]. Phosphorylation of MYC at serine residue 62 is induced by ERK downstream of RAS, positively regulating the protein stability of MYC [97]. In contrast, MYC phosphorylation at threonine residue 58 via GSK3 is destabilizing and frequently mutated in B-cell lymphoma to prevent phosphorylation and decay [98,99]. Furthermore, the isomerase PIN1 alters the configuration of proline residue 63, which sterically shields phosphorylated serine 62 from dephosphorylation, eventually protecting MYC from degradation [100]. This explains why KO of PIN1 significantly increased the survival of Eµ-Myc transgenic mice due to lowered MYC levels [101].

**Figure 4 cells-12-00037-f004:**
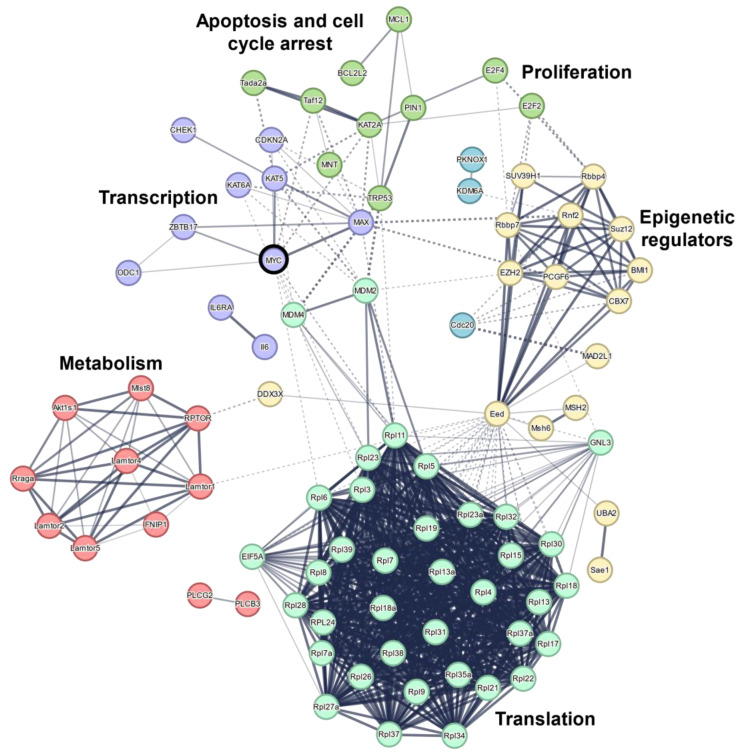
Common effectors of MYC-induced lymphomagenesis physically interact. Critical genes from Figure 2 were clustered based on the physical interaction of the encoding protein (line thickness indicates the strength of data support) of the encoded protein using the STRING database [102]. Additional common interaction partners are shown. Terms describe biological function of some proteins in the clusters.

Recently, we discovered a novel way to target the stability of MYC by exploiting its constant shuttling from the cytoplasm to the nucleus [44]. MYC is well-known for associating with microtubules [103], and the post-translational acetylation of tubulin is a crucial factor for assembly and stability [104,105]. The histone deacetylase 6 (HDAC6) mediates tubulin deacetylation, and pharmacological inhibition of HDAC6 decreases MYC protein levels, likely by heat-shock protein-mediated proteolysis [44]. Similarly, the broader MYC interaction network could be further utilized for therapeutic interventions by targeting factors that connect central pathways, such as MIZ-1 (*Zbtb17*), EED, or DDX3X (Figure 4).

### 4.3. MYC-Induced Apoptosis Might Be Triggered by Transcriptional Stress and DNA Damage

Apoptosis can be seen as the result of an altered balance between anti- and pro-apoptotic effectors. Therefore, KO of genes encoding anti-apoptotic mediators, such as MCL-1 or BCL-w, had a positive effect on the survival of Eµ-Myc transgenic mice because malignant cells could not maintain high levels of anti-apoptotic proteins and initiated apoptosis [106,107,108]. Loss of pro-apoptotic mediators such as BIM or PUMA in lymphoma cells had the opposite effect on survival, as expected [109,110,111]. The interplay of MYC and other TFs was thought to modulate the expression of apoptotic mediators in a stoichiometric way. For example, the promoter of BCL2 is antagonistically controlled by MYC and MIZ-1, implying that this interaction balances apoptosis induction [112,113].

However, a newer hypothesis assumes that “MYC-driven apoptosis results from RNA Polymerase II stalling” and not from direct transcriptional control of apoptotic mediators [114]. Overexpression of MYC leads to replicative stress, which can stall RNA Polymerase II [115,116]. Two phenomena appear to be the primary source of this type of stress: (I) transcription-replication conflicts caused by the crashing of replication and transcription machinery during the S phase due to fast replication and high transcriptional output [117]. (II) R-loops, which are transient DNA-RNA hybrids formed by nascent RNA transcripts and act as a barrier to replication fork progression [118,119]. Both stressors may generate DNA lesions and genomic instability, which trigger the DNA damage repair machinery.

In particular, the DNA damage sensor CHK1 seems crucial in MYC-induced lymphomagenesis, as B-cell-specific heterozygous KO completely abrogated malignant transformation [120]. CHK1-haploinsuffiency correlated with higher γ-H2A.X levels and PARP cleavage, concluding cell death induction due to unrepaired DNA damage [120]. Activated ATM and high γ-H2A.X levels were also found when the acetyltransferase TIP60 (Kat5) was partially knocked out in Eµ-Myc transgenic mice [121]. Interestingly, TIP60 modulates DNA damage pathways and can modify MYC post-translationally, increasing MYC’s protein stability [122].

A decrease in active, phosphorylated CHK1 was found in Eµ-Myc lymphomas lacking functional MIZ-1, indicating an impaired activation of DNA damage response [50]. Other interactors of MYC, such as BRCA1, might be involved in resolving R-loops and, thus, protect from tumorigenesis [123,124]. A functional DNA damage response might be necessary to establish MYC-dependent tumors because apoptosis induction is prevented. However, more experimental evidence might be required to fully understand the connection between MYC, transcriptional stress, and DNA damage repair pathways.

### 4.4. Why Is Targeting MYC So Effective but Difficult to Realize?

MYC overexpression is related to many hallmarks of cancer, ranging from immunosuppression to metabolic and epigenetic reprogramming [1]. Importantly, MYC-overexpressing cancer cells show an “oncogene addiction” to MYC, which implies that targeted inactivation leads to tumor regression [125]. Selective pressure, however, enforces mechanisms such as post-translational modifications or mutations to sustain high MYC levels, which eventually results in tumor relapse [125]. It appears evident that targeting MYC in cancer cells would reverse all oncogene-induced effects, restore physiological cell state, or induce apoptosis. Even relapse might be circumvented when MYC levels are entirely eradicated. However, all cells need physiological levels of MYC or other members of the MYC family. Here, the factors controlling MYC stability and translocation into the nucleus are the key to finding new pharmacological targets.

The proteasomal degradation of MYC can be initiated, for example, by constant HDAC6 inhibition [44]. A reduction by half was sufficient to induce apoptosis and cell cycle arrest in B-NHL cells but did not result in lymphoma remission in mice [44]. A critical factor for successful therapy is the immunological visibility of the tumor. As MYC-driven cancers suppress the autocrine secretion of factors mediating senescence or immune cell invasion [126,127], combining the insights from targeting MYC at the protein level and activating the immune surveillance could define future therapies. Still, the impact of epigenetic reprogramming in MYC-induced lymphoma is unknown regarding persistent changes in gene expression, even after uncoupling from the oncogene MYC.

Another approach to prevent MYC-induced malignant transformation would be to target essential factors involved in this process. Notably, all 48 critical genes for murine MYC-induced lymphomagenesis are expressed in human lymphoid tissue and most are deregulated in the corresponding tumor (Figure 5A). In addition, some genes are frequently mutated in human B-NHL (Figure 5B), leading to significantly reduced progression-free survival when mutated (Figure 5C). Further research is needed to address how mouse model findings can be used to prevent or treat human disease.

## 5. Outlook

The mechanisms of how MYC drives malignant transformation are well studied. However, there are still under-investigated areas in MYC-induced lymphomagenesis, such as the role of the three-dimensional chromatin organization or the fatty acid metabolism. Studying the chromatin organization beyond the levels of nucleosomes might further explain the origin of the characteristic *MYC* translocations found in human B-cell lymphoma [131]. First, chromosome loop anchors are fragile sites for genetic rearrangements in B-cells [132]. Second, removing insulators between strong enhancers and oncogenes might allow the ectopic oncogene expression observed in cancer [133]. Third, the increased frequency of R-loops that occur directly at the *MYC* locus was also associated with *MYC* translocations [134].

MYC overexpression in cancer was accompanied by remodeling of the glycerophospholipid metabolism [135,136]. For example, loss of the lipoxygenase ALOX12 dramatically decreased the survival of Eµ-Myc transgenic mice [137]. On the contrary, the loss of one *Myc* allele extended the lifespan of mice characterized by a healthier lipid metabolism [138]. We recently discovered that MYC-induced lymphomagenesis increased the levels of certain polyunsaturated fatty acids, which was associated with elevated mTORC1 activity and impaired autophagy [30]. It would be interesting to test if interfering with lipid remodeling could prevent MYC-induced lymphomagenesis.

At last, the combination of an established MYC-driven cancer mouse model, such as Eµ-Myc (Table 1), with modern-day technologies, such as spatial single-cell sequencing, might be helpful to resolve intercellular differences in *MYC* expression tracking the acquisition of secondary mutations in vivo and to clarify the role of MYC in the tumor microenvironment All these efforts will eventually be rewarded with a deeper understanding of MYC-dependent lymphomagenesis, pointing toward future therapies.

**Table 1 cells-12-00037-t001:** List of all mouse models analyzed combined with the Eµ-Myc transgene and their respective survival.

Name	Gene	Function	Model	Survival	[%]	Ref.
µMT (IgM heavy chain)	*Ighm*	Receptor	Full KO	* CTRL: 120 d, KO: 80 d	66.67	[45]
4E-BP1	*Eif4ebp1*	Translation	Dox-inducible KO	* CTRL: 90 d, KO: 145 d	161.1	[139]
A1/BFL-1	*Bcl2a1a*	Apoptosis	(a) Full KO (b) Transplantation of tamoxifen-inducible KO cells (c) Constitutive miR-shRNA (KD)	(a) CTRL: 92 d, KO: 94 d (b) Vehicle: 17 d, Tamoxifen: 23 d (c) CTRL: 103 d, KD: 109 d	(a) 102.2 (b) 135.3 (c) 105.8	[140,141]
AID	*Aicda*	DNA damage and repair	Full KO	(a) no effect (b) CTRL: 112 d, KO: 130 d	(a) 100 (b) 116.1	[140,141]
ALOX12	*Alox12*	Metabolism	Full heterozygous KO	CTRL: 220 d, +/−: 70 d	31.8	[137]
AMD1	*Amd1*	Metabolism	Transplanted shRNA transduced FL cells	* CTRL: 112 d, KO: 70 d	62.5	[142]
EIF5A	*Eif5a*	Translation	Transplanted shRNA transduced FL cells	* CTRL: 112 d, KO: 56 d	50	[142]
AMPKα1	*Prkaa1*	Signaling	Full KO	CTRL: 10 wks, KO: 7 wks	70	[51]
APAF1	*Apaf1*	Apoptosis	Transplanted FL cells from Eµ-Myc full KO mice	No effect	100	[143]
ATF2	*Atf2*	Transcription factor	CD19-cre, B-cell-specific KO	No effect	100	[144]
ATF4	*Atf4*	Transcription factor	Tamoxifen-inducible	* Vehicle: 40 d, Tamoxifen: 80 d	200	[145]
ATF7	*Atf7*	Transcription factor	CD19-cre, B-cell-specific KO	WT: 105 d, KO: 135 d	128.6	[144]
BAD	*Bad*	Apoptosis	Full KO	CTRL: 138 d, WT: 78 d	56.5	[146]
BAX	*Bax*	Apoptosis	Full KO	CTRL: 21.7 wks WT: 12.6 wks	58.1	[146]
BCL-2	*Bcl2*	Apoptosis	(a) Full heterozygous KO (b) Transplanted FL cells from Eµ-Myc full KO mice	(a) CTRL: 116 d, KO: 154 d (b) No effect	(a) 132.8 (b) 100	[147,148]
BCL-W	*Bcl2l2*	Apoptosis	Full KO	CTRL: 90 d, KO: 298.5 s	331.7	[106]
BCL-x	*Bcl2l1*	Apoptosis	(a) Full heterozygous KO (b) Transplantation of tamoxifen-inducible KO cells	(a) CTRL: 116 d, KO: 174 d (b) CTRL: 19 d, KO: 25 d	(a) 150 (b) 131.6	[107,148]
BIF-1	*Sh3glb1*	Apoptosis	Full KO	CTRL: 107 d, KO: 65 d	60.7	[149]
BIK	*Bik*	Apoptosis	Full KO	No effect	100	[150]
BIM	*Bcl2l11*	Apoptosis	(a) Full KO (b) Mb1-cre, B-cell-specific KO	(a) CTRL: 15 wks, KO: 8.2 wk (b) CTRL: 72 d, KO: 113 d	(a) 54.7 (b) 63.7	[109,110]
BMF	*Bmf*	Apoptosis	Full KO	CTRL: 138 d, KO: 87 d	63	[151]
BMI1	*Bmi1*	Epigenetic regulator	(a,b) Full heterozygous KO (c) Transplanted overexpressing FL cells	(a) * CTRL: 150 d, KO: >300 d (b) * CTRL: 100 d, KO: >250 d (c) CTRL: >300 d, OE: 74 d	(a) >200 (b) >250 (c) 24.7	[152,153,154]
BOK	*Bok*	Apoptosis	Full KO	CTRL: 107 d; KO: 121 d	113.1	[155]
BTK/TEC	*Btk* *Tec*	Signaling	Full heterozygous KO: BTK+/− TEC+/−	CTRL: 100 d, KO: 60 d	60	[156]
BUB1	*Bub1*	PTM	Overexpression of point mutant (T85)	CTRL: 21 wks, MUT: 13 wks	61.9	[157]
CAML	*Caml*	Signaling	Subcutaneous transplant of tamoxifen-inducible full KO	Vehicle: 7 d, Tamoxifen: >25 d	>357	[158]
Caspase 9	*Casp9*	Apoptosis	FL transplantation of full KO cells	CTRL: 57 wk; KO: 54 wk	94.7	[143]
Caspase 2	*Casp2*	Apoptosis	Full KO	CTRL: 16 wks, KO: 8 wks	50	[159]
CBX7	*Cbx7*	Epigenetic regulator	FL cells with overexpression	CTRL: >300 d, OE: 43 d	<14.3	[154]
CD19	*Cd19*	Receptor	Full KO	CTRL: 13.4 wks, KO: 24.3 wks	181.3	[43]
CDK4	*Cdk4*	pRB-axis	Full KO	CTRL: 18 wks, KO: 11 wks	61.1	[160]
CHK1	*Chek1*	DNA damage and repair	(a) Full heterozygous KO (b) Mb1-cre, B-cell-specific KO	(a) CTRL: 106 d, KO: 205 d (b) CTRL: 106 d, KO: >350 d	(a) 193 (b) >330	[120]
CKS1	*Cks1b*	pRB-axis	Full KO	CTRL: 91 d, KO: 268 d	294.5	[161]
CREBBP	*Crebbp*	Epigenetic regulator	AID-cre + immunization	* CTRL: 85 d, KO: 55 d	64.7	[162]
cREL	*Rel*	Transcription factor	Full KO	CTRL: 115 d, KO: 79 d	68.7	[163]
CSN6	*Cops6*	PTM	Full heterozygous KO	* CTRL: 100 d, KO: 190 d	190	[164]
CUL9	*Cul9*	PTM	Full KO	CTRL: 126.4 d, KO: 85.1 d	67.3	[165]
DDX3X	*Ddx3x*	Helicase	(a) CD19-cre, B-cell-specific KO (b) Vav-cre, B-cell-specific KO	(a) ♂: CTRL: 83 d, KO: 105 d; ♀: CTRL: 87 d, KO: 212 d (b) ♂: CTRL: 98 d, KO: >350 d ♀: CTRL: 110.5 d; KO: 83 d	(a)♂: 126.5; ♀: 243.7 (b)♂: >357.1; ♀: 75.1	[62]
DICER	*Dicer1*	Splicing	CD19-cre, B-cell-specific KO	WT: 194 d, KO: 351 d	180.9	[166]
DMP1	*Dmp1*	p53-axis	Full KO	* CTRL: 22 wks, KO: 13 wks	59.1	[167]
DNMT3B	*Dnmt3b*	Epigenetic regulator	Full heterozygous KO	* CTRL: 125 d, KO: 75 d	60	[168]
DPY30	*Dpy30*	Epigenetic regulator	Full heterozygous KO	CTRL: 121 d, KO: 180.5 d	149.2	[169]
E2F1	*E2f1*	pRB-axis	(a,b) Full KO	(a) * CTRL: 24 wks, KO: 16 wks (b) No effect	(a) 150 (b) 100	[78,170]
E2F2	*E2f2*	pRB-axis	Full KO	WT: 126 d, KO: 63 d	50	[78]
E2F3	*E2f3*	pRB-axis	Full KO	No effect	100	[78]
E2F4	*E2f4*	pRB-axis	Full KO	CTRL: 110 d, KO: 375 d	340.9	[78]
E6AP	*Ube3a*	PTM	Full heterozygous KO	CTRL: 103 d, KO: 153 d	148.5	[52]
EZH2	*Ezh2*	Epigenetic regulator	(a) GOF mutant (b) Transplanted shRNA transduced FL cells	(a) CTRL: 137.5 d, MUT: 51 d (b) CTRL: 220 d, KD:55 d	(a) 37.1 (b) 25	[86,153]
FNIP1	*Fnip1*	Metabolism	Full KO	* CTRL: 110 d, KO: >300 d	>272.7	[171]
FOXO	*Foxo4*	TF	Dominant negative mutant, transplanted transduced FL cells	* CTRL: >250 d, MUT: 50 d	<20	[172]
GCN2	*Eif2ak4*	Translation	Transplanted tamoxifen-inducible lymphoma cells	No effect	100	[145]
GCN5	*Kat2a*	Epigenetic regulator	CD19-cre, B-cell-specific KO	CTRL: 21 wks, KO: 58.4 wks	278.1	[83]
H2A.X	*H2ax*	Epigenetic regulator	Full KO	No effect	100	[25]
HDAC1	*Hdac1*	Epigenetic regulator	Mb1-cre, B-cell-specific KO	CTRL: 161 d, KO: 170 d	105.6	[173]
HDAC2	*Hdac2*	Epigenetic regulator	Mb1-cre, B-cell-specific KO	CTRL: 161 d, KO: 164 d	101.9	[173]
IBTK	*Ibtk*	Signaling	Full KO	CTRL: 90 d, KO: 150 d	166.7	[174]
ID2	*Id2*	TF	Full KO	No effect	100	[175]
IL6R (gp130)	*Il6ra*	Receptor	FL xenograft with CD19-cre deleted cells	CTRL: 277 d, KO: 20 d	7.2	[176]
IL7R	*Il7r*	Receptor	(a) LOF (no activation of survival mechanism) (b) Transplanted cells	(a) CTRL: 15.5 wks, MUT: 66.5 wks (b) No effect	(a) 429 (b) 100	[177]
INK4A/P16	*Cdkn2a*	p53-axis	Full heterozygous KO	* CTRL: 150 d, KO: 45 d	30	[152]
INK4C/P18	*Cdkn2c*	p53-axis	Full KO	No effect	100	[175]
KLRK1	*Klrk1*	Receptor	Full KO	WT: 22 wks, KO: 15 wks	68.2	[178]
KSR1	*Ksr1*	Signaling	Full KO	CTRL: 95 d, KO: 138 d	145.3	[179]
L24	*Rpl24*	Translation	Full heterozygous KO	* CTRL: 100 d, KO: 210 d	210	[180]
L38	*Rpl38*	Epigenetic regulator	Full heterozygous KO	* CTRL: 70 d, KO: 110 d	157.1	[180]
LGL	*Llgl1*	Cytoskeleton	Full KO	No effect	100	[181]
MAD2	*Mad2l1*	Spindle assembly	Transplanted HSCs with overexpression	* CTRL: >350 d, OE: 60 d	<17.1	[182]
MAX	*Max*	TF	Mb1-cre, B-cell-specific KO	CTRL: 97 d, KO: 300 d	309.3	[92]
MCL1	*Mcl1*	Apoptosis	(a) CD19-cre, B-cell-specific KO (b) Rag1-cre, heterozygous KO (c) Transplanted tamoxifen-inducible lymphoma cells (d) Transgene (H2K promoter) (e) Transgene (VavP promoter)	(a) CTRL: 91 d, KO: 123 d (b) CTRL: 129 d, KO: 346 d (c) WT: 19 d, KO: 35 d (d) WT: 134 d, OE: 72 d (e) WT: 94 d, OE: 30.5 d	(a) 135.2 (b) 268.2 (c) 184.2 (d) 53.7 (e) 32.4	[107,108,183,184]
MDM2	*Mdm2*	p53-axis	(a) Full heterozygous KO (b) Point mutation (LOF) C305F	(a) CTRL: 20.6 wks, KO: 44.3 wks (b) CTRL: 20.7 wks MUT: 11.6 wks	(a) 215 (b) 56.0	[69,74]
MDM4	*Mdm4*	p53-axis	(a) Transgene (b) Full heterozygous KO	(a) CTRL: 31 wks, OE: 34 wks (b) * CTRL: 350 d, KO: >400 d	109.7 >114.3	[185,186]
MDMX	*Mdmx*	Deleted in mice	Point mutation W201S/W202G	* CTRL: 170 d, MUT: 80 d	47.1	[73]
MGA	*Mga*	TF	CD19-cre, B-cell-specific KO	CTRL: 97 d, KO: 87 d	89.7	[187]
MHCII	*H2*	Receptor	Full KO + immunization	No effect	100	[162]
MIF	*Mif*	Cytokine	Full KO	CTRL: 2.67 months, KO: 3.67 months	137.5	[188]
miR146a	*Mir146*	microRNA	Full KO	CTRL: 104.5 d, KO: 82.5 d	78.9	[189]
miR-17-92	*Mir17hg*	microRNA	(a) Transplanted overexpressing FL cells (b) Transplanted tamoxifen-inducible KO cells	(a) * CTRL: >200 d, OE: 125 d (b) * CTRL: 20 d, KO: 33 d	<62.5 165	[190,191]
MIZ-1	*Zbtb17*	TF	Mb1-cre, B-cell-specific KO	* CTRL: 110 d, KO: 350 d	318.2	[50]
MNT	*Mnt*	TF	(a) Full heterozygous KO (b) Rag1-cre, KO	(a) CTRL: 17 wks KO: 28 wks (b) CTRL: 86 d, KO: 463 d	(a) 538.4 (b) 164.7	[192,193]
MOZ	*Kat6a*	Epigenetic regulator	Full heterozygous KO	CTRL: 105 d, KO: 411 d	391.4	[84]
MPL	*Mpl*	Receptor	Full KO	CTRL: 87 d, KO: 76.5	87.9	[194]
MSH2	*Msh2*	DNA damage and repair	(a) Full KO (b) Mutation (G674A)	(a) * CTRL: 100 d, KO: 40 d (b) * CTRL: 100 d, MUT: 40 d	(a) 40 (b) 40	[45]
MTAP	*Mtap*	Metabolism	Full heterozygous KO	CTRL: 130 d, KO: 87 d	66.9	[195]
MTBP	*Mtbp*	p53-axis	Full heterozygous KO	CTRL: 135 d, KO: 270 d	200	[196]
MYSM1	*Mysm1*	PTM	Tamoxifen-inducible full KO	* CTRL: >150 d, KO: 80 d	<53.3	[197]
NFKB1/P105	*Nfkb1*	TF	Full KO	No effect	100	[198]
NFKB2/P100	*Nfkb2*	TF	Full KO	CTRL: 205 d, KO: 171 d	83.4	[199]
NOXA	*Pmaip1*	Apoptosis	Full KO	No effect	100	[111]
Nucleostemin	*Gnl3*	Signaling	Full heterozygous KO	* CTRL: 100 d, KO: 260 d	260	[200]
ODC	*Odc1*	Metabolism	Full heterozygous KO	CTRL: 110 d, KO: 320 d	290.9	[201]
OGG1	*Ogg1*	DNA damage and repair	Full KO	No effect	100	[202]
p19/ARF	*Cdkn2a*	p53-axis	(a) Full heterozygous KO (b) Full KO (c) Full KO	(a) * CTRL:135 d, KO: 35 d (b) CTRL: 20.7 wks, KO: 10.1 wks (c) CTRL: 89 d, KO: 73 d	(a) 25.9 (b) 48.8 (c) 82.0	[71,72,74]
p27	*Cdkn1b*	p53-axis	Full KO	CTRL: 120 d, KO: 80 d	66.7	[203]
p38	*Mapk14*	Signaling	Heterozygous mutation (T180A, T182F)	CTRL: 77 d, MUT: 85	110.4	[204]
p53	*Trp53*	p53-axis	(a) Full heterozygous KO (b) Full heterozygous KO (c) Full heterozygous KO (d) Full KO (e) Full heterozygous KO (f) Point mutation LOF (G515C)	(a) CTRL: 20.6 wks, KO: 5.6 wks (b) * CTRL: 137.5 d, KO: 37.5 d (c) * CTRL: 100 d, KO: 30 d (d) CTRL: 89 d, KO: 40 d (e) CTRL: 138 d, KO: 35 d (f) CTRL: 138 d, MUT: 62 d	(a) 27.2 (b) 27.3 (c) 30 (d) 44.9 (e) 25.4 (f) 44.9	[69,70,71,72,111]
P73	*Trp73*	TF	Full KO	No effect	100	[205]
PARP1	*Parp1*	Apoptosis	Full KO	CTRL: 127 d, KO: 90 d	70.1	[206]
PARP2	*Parp2*	Apoptosis	Full KO	CTRL: 127 d, KO: 326 d	257	[206]
PARP14	*Parp14*	Metabolism	Full KO	* CTRL: 13 wks, KO: 20 wks	153.8	[207]
PCGF6	*Pcgf6*	Epigenetic regulator	CD19-cre, B-cell-specific KO	CTRL: 203 d, KO: 65 d	32.0	[187]
PFP	*Prf1*	Apoptosis	Full KO	CTRL: 135 d, KO: 139 d	103	[208]
PIN1	*Pin1*	Isomerase	Full KO	CTRL: 108 d, KO: 431 d	399.1	[101]
PLCβ3	*Plcb3*	Signaling	Full heterozygous KO	* CTRL: >365 d, KO: 100 d	<27.4	[209]
PLCγ2	*Plcg2*	Signaling	Full KO	* CTRL 20 wks, KO: 10 wks	50	[210]
PML	*Pml*	Apoptosis, Signaling	Full heterozygous KO	CTRL: 103 d, KO: 153 d	149	[52]
PRDM11	*Prdm11*	Epigenetic regulator	Full KO	CTRL: 113 d, KO: 94 d	83.2	[211]
PRDM15	*Prdm15*	Transcription	Tamoxifen-inducible KO	CTRL: 107 d, KO: 332 d	310.3	[212]
PREP1	*Pknox1*	TF	Tamoxifen-inducible heterozygous KO	CTRL: 58 wks, KO: 23 wks	39.7	[213]
PRMT5	*Prmt5*	RNA/Splicing	Tamoxifen-inducible heterozygous KO	* CTRL: 90 d, KO: 175 d	194.4	[17]
PUMA	*Bbc3*	Apoptosis	(a) Full KO (b) Full KO	(a) CTRL: 100 d, KO: 66 d (b) * CTRL: 15 wks, KO: 11 wks	(a) 66 (b) 73.3	[111,214]
RAC1	*Rac1*	Signaling	Transplanted, transduced cells	* CTRL: 18 d, KD: 28 d	155.6	[215]
RAG1	*Rag1*	DNA damage and repair	Full KO	* CTRL 110 d, KO: 90 d	81.8	[141]
RAIDD	*Cradd*	Apoptosis	Full KO	* CTRL: 120 d, KO: 110	91.7	[216]
RAP1	*Terf2ip*	Signaling	Full KO	* CTRL: 15 wks, KO: 12 wks	80	[217]
RAPTOR	*Rptor*	Metabolism	CD2-cre	* CTRL: 18 wks, KO: >55 wks	>305.6	[218]
pRB	*Rb1*	pRB-axis	Full heterozygous KO	* CTRL: 135 d, KO: 125 d	92.6	[71]
RIPK3	*Ripk3*	Signaling	Full KO	CTRL: 118 d, KO: 97 d	82.2	[219]
RUNX1	*Runx1*	TF	Mx1-cre + pIpC; heterozygous for p53	No effect	100	[220]
SAE2	*Uba2*	PTM	Transplanted, transduced lymphoma cells	* CTRL: 35 d, KD: >100 d	>285.7	[221]
Scribble	*Scrib*	Scaffold	Transplanted FL cells	* CTRL: 175 d, KO: 280 d	160	[222]
Septin 4	*Septin4*	Cytoskeleton	Full KO	* CTRL: 270 d, KO: 100 d	37.0	[223]
Sirtuin 4	*Sirt4*	Epigenetic regulator	Full KO	CTRL: 195 d, KO: 139 d	71.3	[224]
SKP2	*Skp2*	PTM	Full KO	CTRL: 100 d, KO: 150 d	150	[225]
SMARCAL1	*Smarcal1*	Helicase	Full KO	CTRL: 187 d, KO: 224 d	119.8	[116]
SMYD2	*Smyd2*	Epigenetic regulator	CD19-cre, B-cell-specific KO	* CTRL: 150 d, KO: 175 d	116.7	[226]
SUV39H1	*Suv39h1*	Epigenetic regulator	Full KO	* CTRL: 125 d, KO: 60 d	48	[90]
SUZ12	*Suz12*	Epigenetic regulator	Heterozygous LOF mutation	CTRL: 103 d, MUT: 72 d	69.9	[153]
TCRα	*Trac*	Receptor	Full KO	No effect	100	[45]
TCRΔ	*Trdc*	Receptor	Full KO	No effect	100	[45]
TEL2	*Etv7*	Deleted in mice	Transplanted, transduced cells (overexpression)	CTRL: >16 wks, OE: 13 wks	81.3	[227]
TIP60	*Kat5*	Epigenetic regulator	Full heterozygous KO	* CTRL: 52 wks, KO: 12 wks	23.1	[121]
TIS11B	*Zfp36l1*	Transcription	Eµ-Tis11b (overexpression)	* CTRL: 140 d, OE: 100 d	71.4	[228]
TRAIL-R	*Tnfrsf10b*	Apoptosis	Full KO	CTRL: 119 d, KO: 82 d	68.9	[229]
Tristetraprolin	*Zfp36*	Transcription	Eµ-TTP (overexpression)	(a) CTRL: 103.5 d, OE: 194 d (b) CTRL: 121 d, OE: 277 d	(a) 187.4 (b) 228.9	[228]
UCH-L1	*Uchl1*	Ubiquitin system	(a) Transgene (b) Full KO	(a) * CTRL: >60 wks, TG: 45 wks (b) * CTRL: >60 wks, KO: >60 wks	(a) <75 (b) 100	[230]
UNG1	*Ung*	Repair	Full KO	* CTRL: 110 d, KO: 85 d	77.3	[202]
UTX	*Kdm6a*	Epigenetic regulator	CD19-cre, B-cell-specific KO	* ♂: CTRL: 145 d, KO: 120 d; ♀: CTRL: >200 d, KO: 70 d	♂: 82.8 ♀: <35	[231]
WIP1	*Ppm1d*	Signaling	Full KO	CTRL: 77 d, KO: 138 d	179.2	[204]
WRN	*Wrn*	Helicase	Mutation in helicase domain	CTRL: 115 d, KO: 151 d	131.3	[232]
XPO1	*Xpo1*	Nuclear export	Point mutation (E571K), tamoxifen-inducible	CTRL: 35 d, KO: 28 d	80	[233]
ZMAT3	*Zmat3*	Transcription	Full KO	CTRL: 125 d, KO: 93 d	74.4	[234]
ZRANB3	*Zranb3*	Helicase	Full KO	CTRL: 104 d, KO: 138 d	132.7	[116]

Relative survival was calculated based on the control (CTRL) cohort. A symbol (*) designates estimated median survival based on the presented survival curve.

## Figures and Tables

**Figure 1 cells-12-00037-f001:**
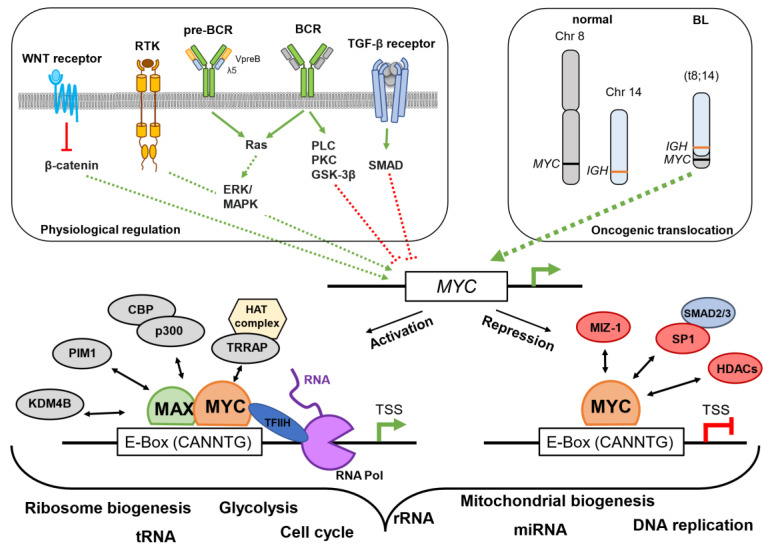
Regulation of the *MYC* gene and functions of MYC. Shown are pathways that activate or repress *MYC* transcription. Oncogenic translocations, such as (t8;14) found in human BL, juxtapose *MYC* and potent enhancers, resulting in high *MYC* expression. MYC has many interaction partners, leading to context-dependent regulation of biological processes. This scheme summarizes findings from various sources [2,8,20,21,22]. BCR—B-cell receptor.

**Figure 2 cells-12-00037-f002:**
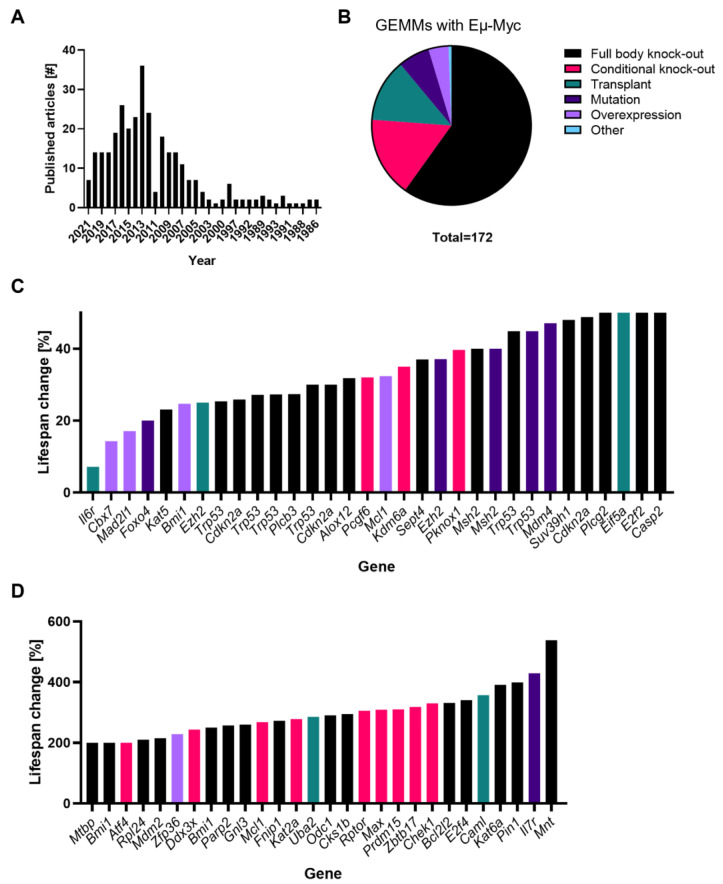
A meta-analysis of GEMMs of MYC-induced lymphoma. (**A**) Timeline of published articles using Eµ-Myc transgenic mice found on Pubmed with search terms: “Emu-myc”, “Eµ-myc”, or “Myc-induced lymphoma”. (**B**) Pie chart showing the type of mouse model used. Transplants do not fall under the term “GEMM”. (**C**,**D**) Ranked list of genes that decrease (**C**) or increase (**D**) survival, normalized to the control cohort for each study. Please refer to Table 1 for a complete list of genes.

**Figure 3 cells-12-00037-f003:**
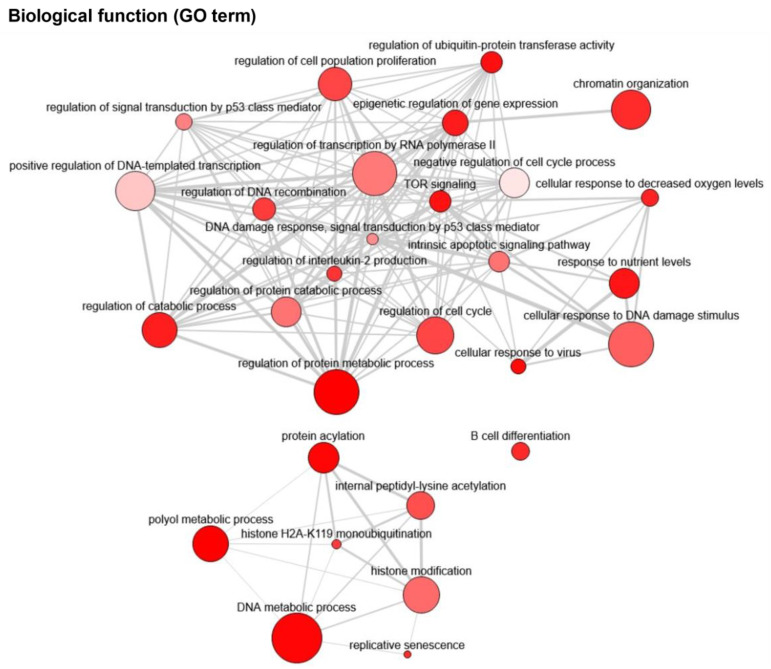
Common biological functions among the critical genes from Figure 2 for Eµ-Myc lymphomas were analyzed using Enrichr and the corresponding GO (gene ontology) terms [66,67]. Visualization was performed using the Revigo tool [68]. The color of the bubble corresponds to the adjusted *p* value (the redder, the lower the *p* value) and the size to the genes under the GO Term.

**Figure 5 cells-12-00037-f005:**
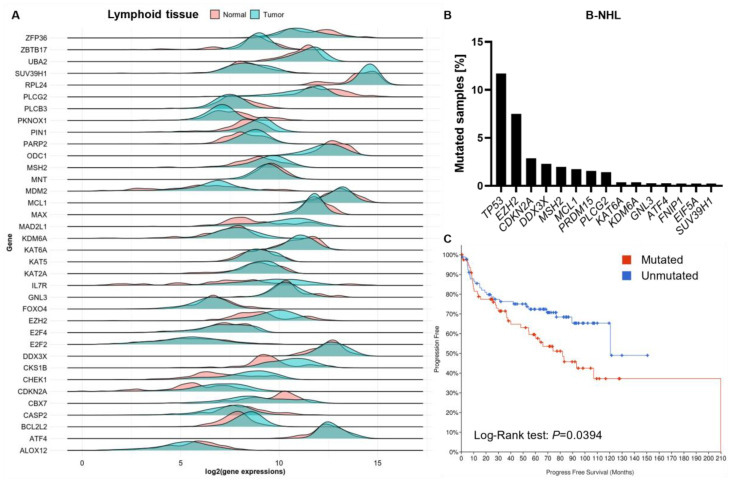
Critical genes for murine MYC-induced lymphoma play a role in human B-NHL. (**A**) Gene expression of the critical genes from Figure 2 is shown for normal and malignant human lymphoid tissue using the TNMplot tool [128]. (**B**) The mutation rate of the critical genes was assessed using cBioPortal [129,130]. A total of 41 out of 48 genes showed mutations in B-NHL. A total of 2117 samples from eight studies were included in the analysis. (**C**) Progression-free survival analysis of B-NHL patients with mutations (n = 81) or no mutations (n = 91) in the 48 critical genes derived from Eµ-Myc mice is shown. Overall survival was unaltered. Data and visualization are derived from cBioPortal [129,130].

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
