# Peer review of "Lessons from Using Genetically Engineered Mouse Models of MYC-Induced Lymphoma"

_cells, 2022, doi:10.3390/cells12010037_

Round 1
Reviewer 1 Report
The authors collected literature data on a vast number papers describing MYC overexpression mouse models to derive relevant data for MYC-induced B-cell lymphoma development in mice and potential treatment.
In the paper the distinction between animal model derived data and human data is not always clear and needs clarification. Is the description in line 43-62 the situation in humans or derived from mouse models? I would guess mainly the latter, by checking the references.
The paper would in addition benefit from a critical appraisal by the authors of the applicability of the model results for humans and the clinical practice. Reference [47] provides a great example how mouse model derived data can be adjusted for human B-NHL
Are the 50 critical genes for MYC-induced lymphomagenesis (line 182) also identified in human MYC overexpressed B-NHL? Is the in 4.1 described interaction of epigenetic modifiers with MYC confirmed in the human situation? And what can be learned from MYC-induced lymphomagenesis for B-NHL with secondary MYC overexpression?
Several sentences require clarification:
-line 110: I don’t understand “however” in this context, please explain
-line 123: I don’t understand “moreover” In this context, do you mean “in addition”?
-line 139: I don’t understand “however” in this context: this seems to be not a contradiction but rather a confirmation?
- lines 146/147 cite reference [6] as “describing more than 4000 differentially expressed genes between individual B-cell lymphomas”. In fact, the study describes that more than 4000 genes in each tumor are differentially expressed relative to control.
-Line 148: please explain why “Thus, gene expression signatures similar to BL or DLBCL can be identified”? Do you suggest that they are thus totally aspecific?
-line 148-153: reference 47 used the mouse model to simulate DLBCL response to therapy . GEP data of the mice were compared to carefully selected genetic subtypes of DLBCL. Therefor the statement “Surprisingly, the genomic signatures of murine and human DLBCL were so similar that……” is not appropriate.
-line 155; please explain “especially considering the expanding sub-classification of B- cell lymphomas.”
General questions and comments:
--line 22: the statement “Overexpression of MYC, as observed in the majority of human malignancies” requires a reference
- line 27: No only MYC translocations lead to overexpression in B-NHL
-line 69: “if” should be “whether”
Line 91: “B220low” cells is missing
Reviewer 2 Report
This is a very well written review. I do not have any suggestions for the existing text although i would suggest
1. some more figures. this would really help the paper clarity. Perhaps expanding on a) the transcription factor co-factors, b) specific points of interactions with key pathways (i.e T cell development, other key pathways)
2. a short paragraph on examples of current/future drug targeting attempts
Round 2
Reviewer 1 Report
all questions were adequately answered